# The Impact of Exotic *Tamarix* Species on Riparian Plant Biodiversity

**Kgalalelo Tshimologo Annie Setshedi [1,\*] and Solomon Wakshom Newete [1,2]** 

[1] School of Animal, Plant and Environmental Sciences, University of Witwatersrand, Private Bag X3, Johannesburg 2050, South Africa; solomonnewete@gmail.com

[2] Agricultural Research Council-Soil, Climate and Water (ARC-SCW), Geo-Information Science Division, Private Bag X79, Arcadia, Pretoria 0001, South Africa

\* Correspondence: Kgalalelosetshedi@gmail.com

**Abstract:** This study investigated the impact of exotic *Tamarix* species on vascular plant biodiversity in riparian ecosystems in the Western Cape Province, South Africa. Vegetation was sampled, using 5 m wide belt transects, along the Leeu, Swart, and Olifants riparian areas, which had varying invasion intensities. Each transect was split into three zones (Zone 1: 0–15 m; Zone 2: 15–35, and Zone 3: >35 m), which were identified at each site based on species composition across each riparian zone. Woody plant species were identified, counted, and their heights measured within the transects that were laid out from the waterpoint (Zone 1) outwards (Zone 2 and 3). Herbaceous aerial cover (HAC) was determined subjectively and objectified using the Walker aerial cover scale. Leeu River had the highest species richness (Dmg = 2.79), diversity (H′ = 2.17; −lnλ = 1.91; N1 = 8.76 and α = 4.13), and evenness (J′ = 0.80). The Swart River had the lowest species richness, which declined from Dmg = 1.96 (Zone 1) to Dmg = 1.82 (Zone 3). Exotic *Tamarix* species ranked in the top three most abundant woody vascular plant species along the Swart and Olifants rivers, where they ranked first and third, respectively. The Jaccard's and Sorenson's coefficients of similarity indicated that species differed greatly between the different sites, $\overline{x} < 27\%$ for both indices. The indices also indicated that the Swart River had the lowest level of species distinctness between zones ($\overline{x} > 80\%$) while the Leeu River had the highest level of species distinctness ($\overline{x} < 50\%$) between the different zones. These findings suggest a possible displacement of herbaceous and woody tree species by exotic *Tamarix* invasion, inter alia, a decrease in ecosystem functions and services associated with the loss in biodiversity, as well as significant bearings on the agricultural ecosystem by reducing the faunal diversity such as crop pollinators, inter alia.

**Keywords:** exotic *Tamarix*; riparian zone; biodiversity; richness; diversity; evenness; biodiversity indicators

## 1. Introduction

The major causes of biodiversity loss are believed to be attributed to direct habitat destruction, invasive organisms, pollution, population growth, and over-exploitation/over-harvesting of natural resources [1–3]. All these causes are better known as the 'HIPPO' acronym [4]. Global warming due to climate change has also been identified as one of the factors that cause the loss of biodiversity, consequently endangering multiple species that are not adapted to a wide range of weather conditions [5]. The loss of plant biodiversity is directly linked to a loss in ecosystem functions, maintenance, resilience, and ecosystem services [6]. Ecosystem functions are ecological processes that control energy, nutrient fluxes, and organic matter fluxes, whilst ecosystem services are benefits that are provided to humanity by the interaction between biotic and abiotic components of the environment [7].

The lack of plant conservation directly cascades into the loss of fauna and entomofauna [8]. Exotic *Tamarix* species have been identified as riparian invaders and are well studied in the United States of America (USA) [9]. Several studies conducted in South Africa have identified riparian zones as regions highly invaded by exotic *Tamarix* species [10–13]. Invasive alien plants (IAPs) not only change the plant species composition and distribution patterns but they alter litter quantity and quality. Nutrient cycling regimes change as a result of invasion by alien plants. Furthermore, ground cover which provides surface stability is lost; this eventually leads to an increase in soil erosion [12,14]. Invasive species can alter fire regimes to favour the establishment of alien plants. Catchment hydrology, sediment yield, and geomorphology are also altered upon invasion [12]. Native riparian vegetation and ecosystems have been modified extensively, and there is an urgent need for management and restoration strategies.

Pyšek and Richardson [15] identified key traits associated with the invasive potential of many alien plants that give them a competitive advantage over indigenous plant species by comparing 64 IAPs. Downey and Richardson [16] defined six thresholds along the extinction trajectory and found that, although no plants are extinct in the wild or globally due to alien plant invasions, native plants have crossed other thresholds along the extinction trajectory due to alien plant invasions. The decline in native plant populations and extirpations due to exotic *Tamarix* invasion in South Africa is yet to be quantified. Newete et al. [13] compared the density of exotic *Tamarix* species to the co-occurring plants and found that the *Tamarix* density was greater in seven of the 11 sites investigated.

The plant diversity of riparian communities provides shelter and food to animal populations, thus, promoting integrity and even further growth of overall biodiversity. Maintaining riparian corridors is important for diversity conservation [17]; however, they also provide an introduction pathway for IAPs across landscape patches, which threaten native plant populations. Riparian areas play a critical role with regards to acting as water sources and biodiversity hotspots, and, therefore, need to be managed more-so in South Africa where they have been identified as vulnerable and impacted by anthropogenic activities as well as IAPs [12,17]. Riparian vegetation differs across various biogeographical areas [12]. Therefore, assessments of invasion should be conducted separately for different areas. The determination of biodiversity relies on the assessment of species richness, evenness, and heterogeneity within an ecosystem [7].

Over 10 million ha of land in South Africa is invaded by alien plants [18] and *Tamarix* Ladeb, is one of them. Although the country has an indigenous *Tamarix* species (*T. usneoides*), the two exotic *Tamarix* species (*Tamarix ramosissima* L. and *Tamarix chinensis* L.) have been in the country close to a century after their first introduction reportedly as garden plants for ornamental purposes [19]. The two species, along with their hybrids, are currently threatening many of the riparian ecosystems in the country and they are listed under the National Environmental Management: Biodiversity Act 2014 (NEM: BA) as category 1b invasive weed requiring urgent management intervention [13,20]). Alien *Tamarix* species are also considered as one of the 100 worst invasive species globally, under the International Union for Conservation of Nature (IUCN) list of Global Invasive Species Database [21]. *Tamarix ramosissima* and *T. chinensis* have been found to hybridise between themselves and with the native congeneric, *T. usneoides*, further extending their distribution range across the country [22]. This has accelerated the species' invasive potential, and complicated control of the exotics as the species are cryptic [13,22–25]. Exotic *Tamarix* species are well adapted to survive adverse conditions, i.e., frost, floods, or drought, and they are also able to re-sprout, regenerate, and establish post-natural disasters [20,26].

The Southern African Plant Invaders Atlas (SAPIA) database identified woody species as the most dominant invaders of riparian zones, which prompted this study, to investigate the impact that exotic *Tamarix* species had on vascular plant diversity in riparian ecosystems that had been invaded by the trees. To achieve the aim, species richness, diversity, evenness, and complementarity were determined across sites that had varying *Tamarix* populations in terms of the *Tamarix* species that had been previously identified at those sites. Establishing the species biodiversity would not only fill the knowledge gaps with regards to which ecosystem services and functions are affected as a result of

invasion, but this will eventually lead to improved and better-informed mitigation approaches to protect indigenous plant biodiversity.

## 2. Methods and Materials

### 2.1. Study Site

The study was conducted at three riparian sites in the Western Cape Province, South Africa, where *Tamarix* invasion had been identified as most prevalent [13]. Both Leeu River (S32,76794 and E21,97958), near the town of Leeu Gamka and the Swart River (S33,16627 and E21,97994) near the town of Prince Albert were in the Central Karoo District of Western Cape. The third site, the Olifants River (S33,50776 and E22,69586), was in the Little Karoo (part of the Cape Floristic Region) about 10–15 km away from the town of De Rust, Western Cape. The World Geodetic System (WGS84) geographical reference system was used to obtain the Global Positioning System (GPS) coordinates. The Leeu River was the only site where the native congeneric, *Tamarix usneoides*, occurred. Both the central Karoo and the Little Karoo are semi-arid regions with dwarf succulent shrubs being a common floral feature [27,28]. Seasonal and periodic droughts are also common in the Karoo area [29].

### 2.2. Data Collection

Experimental Design

The study was conducted between late October and early November during the flowering season of most Karoo vegetation. Vegetation was sampled, using 5 m wide belt transects, along three riparian areas at Leeu, Swart, and Olifants rivers. Woody plant species were identified and counted before their heights and stem diameters were measured within the transects that were laid out from the waterpoint outwards (transects varied in numbers: 3–6 belt transect, length: 50–75 m, and width: 25–50 m). Herbaceous aerial cover (HAC) was determined subjectively, as a percentage, within three 1 $m^2$ quadrats laid out systematically within the 25 $m^2$ quadrats and objectified using the Domin scale [30]. Plant growth forms were later classified into several categories, according to Germishuizen and Meyer [31] and African Plant Database [32]. Each transect was split into three zones (Zone 1: 0–15 m; Zone 2: 15–35; and Zone 3: >35 m), which were identified at each site based on species composition across each riparian zone. GPS coordinates (WGS84 geographical reference system) and altitude were recorded at the beginning of each 25 $m^2$ quadrat.

### 2.3. Data Analysis

Determining Species Richness, Plant Diversity, and Evenness

Species accumulation (Sample-based species accumulation) and rarefaction (individual-based species accumulation) curves were plotted for the three sites. The comparisons between both curves provides information on whether the sampled sites are homogeneous or heterogeneous, indicating how plant species differ across the riparian zones. Sample plots were randomised 100 times to compute the mean estimator and expected species richness for each sample plot accumulation level using Estimate S [33].

Species diversity entails richness (the number of species) and evenness (the equality in the number of individuals for every species). The Margalef's index for species richness ($D_{mg}$), Shannon–Weiner diversity index (H'), the Simpson diversity index ($-\ln\lambda$), Hills diversity number (H1), Fisher's alpha diversity index ($\alpha$), and the Shannon evenness index (J') were used to determine richness, diversity, and evenness (Magurran, 1988; Williams et al., 2005). Indices were either directly computed by Estimate S or as parameters that were used as substitutes into equations that were obtained from Estimate S [33]. Rank abundance curves (determine the number of individuals for different species and rank species from most to least abundant) were plotted in order to further analyse the patterns of diversity in terms of richness and evenness.

The Shapiro–Wilk's W test was used to examine the normality of the indices. The Fligner–Killeen test was used to test for homoscedasticity. Two-way ANOVAs (R version 4.0.2) were conducted, for the six indices, to determine whether there were significant differences at the zonal and site levels. One-way ANOVAs and Tukey's Honest Significant Difference (HSD) multiple comparison post hoc tests were also conducted in order to establish whether, and to what extent, the zone and site impacted on the indices.

### 2.4. Complementarity and Similarity

The Jaccard and Sorenson's similarity indices were used to determine how similar or different species composition was between zones sampled at each site. These indices provide a measure of β-diversity (inter-habitat diversity) as well as the species turnover or species composition change along an environmental gradient where environmental dynamics differ. The values were computed on Estimate S and returned as a percentage.

## 3. Results

### 3.1. Species Richness, Diversity, and Evenness

Species Identification: Family Name, Binomial Nomenclature, Plant Type, Origin, and Status

A total of 42 vascular plant species from 32 genera within 15 families were identified and sampled across the Swart, Olifants, and the Leeu rivers (Table 1). The most species-rich families were Aizoaceae (seven species), Poaceae (6), Amaranthaceae (4), and Fabaceae (4). Asteraceae and Solanaceae both comprised of three species each, with the remaining nine families having only one species. The *Galenia* L., *Atriplex* L., *Prosopis* L., *Argemone* L., *Lycium* L., and *Tamarix* genera all had more than one species. Species richness varied between the different rivers with the Swart, Olifants, and Leeu rivers having 13, 25, and 18 vascular plants, respectively. A total of nine alien (not endemic or indigenous to South Africa) plants were recorded across all three riparian areas, excluding exotic *Tamarix* species. Six of the alien plants are known invasive species listed either under the Conservation of Agricultural Resources Act, 1983 (Act No. 43 of 1983) [34] legislation or the National Environmental Management: Biodiversity Act, 2004 (Act No. 10 of 2004) [35] legislation. Plants occurring naturally in South Africa were all listed as Least Concern, according to the South African National Biodiversity Institute (SANBI) Redlist plant species list.

**Table 1.** Plants sampled along the Swart (S), Olifants (O), and Leeu (L) rivers in Western Cape, South Africa, where exotic *Tamarix* invasion was most prevalent. Alien plants have been marked with an * while listed invasive species have been marked with **.

| Family | Species | Plant Type | Origin | Site Occurred | Status |
|---|---|---|---|---|---|
| Aizoaceae | *Galenia africana* L. | Shrub | South Africa | S, O, | |
| Aizoaceae | *Galenia pubescens* L. | Dwarf shrub | South Africa | O | |
| Aizoaceae | *Lampranthus uniflorus* L. Bolus | Succulent-Shrub | South Africa | S | |
| Aizoaceae | *Mesembryanthemum crystallinum* L. | Succulent | South Africa | S, O, L | |
| Aizoaceae | *Prenia tetragona* Thunb. | Succulent | South Africa | | |
| Aizoaceae | *Psilocaulon coriarium* Burch. ex N.E. Br. | Succulent-Shrub | South Africa | S | |
| Aizoaceae | *Tetragonia tetragonioides* Pall. | Shrub | Eastern Asia, Australia, and New Zealand | S, L | ** |
| Amaranthaceae | *Atriplex muelleri* L. | Dwarf shrub | Australia | S, O | * |
| Amaranthaceae | *Atriplex semibaccata* L. | Dwarf shrub | Australlia | S, O, L | ** |
| Amaranthaceae | *Atriplex vestita* L. | Shrub | South Africa | O, L | |
| Amaranthaceae | *Bassia salsoloides* Fenzl | Dwarf shrub | South Africa | O | |
| Amaranthaceae | *Salsola barbata* Aellen | Shrub | South Africa | S | |
| Anacardiaceae | *Searsia pendulina* Jacq. | Tree | South Africa | L | |

**Table 1.** *Cont.*

| Family | Species | Plant Type | Origin | Site Occurred | Status |
|---|---|---|---|---|---|
| Asteraceae | *Chrysocoma ciliata* L. | Shrub | | O | |
| Asteraceae | *Oncosiphon piluliferum* L.f. | Herb | South Africa | O | |
| Asteraceae | *Senecio burchellii* DC. | Shrub/Dwarf shrub | South Africa | O | |
| Brassicaceae | *Sisymbrium* sp. L. | Herb | Naturalised-introduced in South Africa | O | |
| Brassicaceae | Unidentified | Herb | | O | |
| Chenopodiaceae | *Chenopodium murale* L. | Herb | Europe and parts of Asia and northern Africa | L | * |
| Ebenaceae | *Diospyros lycioides* Desf. | Tree/Shrub | South Africa | L | |
| Fabaceae | *Medicago sativa* L. | Herb | Naturalised-introduced in South Africa | O | |
| Fabaceae | *Prosopis hybrid* L. | Tree/Shrub | Mexico, Central and northern South America | L | * |
| Fabaceae | *Prosopis juliflora* L. | Tree/Shrub | Mexico, Central and northern South America | L | ** |
| Fabaceae | *Vachellia karroo* Hayne | Tree/Shrub | South Africa | S, O, L | |
| Juncaceae | *Juncus kraussii* Hochst. | Herb | South Africa | O | |
| Papaveraceae | *Argemone albiflora* Hornem. | Forb/Herb | North America | O | * |
| Papaveraceae | *Argemone mexicana* L. | Forb/Herb | Mexico | O | ** |
| Poaceae | *Cenchrus* sp. | Graminoid | South Africa | S | |
| Poaceae | *Cenchrus ciliaris* L. | Graminoid | South Africa | O, L | |
| Poaceae | *Cynodon dactylon* L. | Graminoid | South Africa | O, L | |
| Poaceae | *Enneapogon desvauxii* P.Beauv. | Graminoid | South Africa | O | |
| Poaceae | *Hordeum murinum* L. | Graminoid | South Africa | O | |
| Poaceae | *Stipagrostis namaquensis* Nees | Graminoid | South Africa | O, L | |
| Santalaceae | *Viscum rotundifolium* L.f. | Hemiparasite | South Africa | S | |
| Scrophulariaceae | *Sutera* sp.Roth | Shrub | South Africa | S | |
| Solanaceae | *Lycium hirsutum* Dunal | Shrub/Dwarf shrub | | S, L | |
| Solanaceae | *Lycium oxycarpum* Dunal | Shrub/Occasional tree | South Africa | O, L | |
| Solanaceae | *Solanum tomentosum* L. | Shrub/Dwarf shrub | South Africa | O | |
| Tamaricaceae | Exotic *Tamarix* (*T. ramossissima* L. and *T. chinensis* L.) | Tree/Shrub | Eurasia | S, O, L | ** |
| Tamaricaceae | *Tamarix usneoides* L. | Tree/Shrub | Eurasia and South Africa | L | |
| Zygophyllaceae | *Zygophyllum retrofractum* Thunb. | Succulent-Shrub | South Africa | S, L | |

## 3.2. Species Accumulation and Rarefaction Curves for Woody Vegetation

The species accumulation and rarefaction curves for the Olifants and Leeu rivers in the first two zones showed heterogeneous species composition, only in the third zone, which was the furthest away from the waterpoint, did the curves suggest a homogenous species composition (Figure 1). Zones were classified as distances across the river bank with similar species composition (i.e., Zone 1: 0–15 m; Zone 2: 15–35; and Zone 3: >35 m). The Swart River species accumulation and rarefaction curves showed that the species composition was homogenous across all the three zones. All the curves reached an asymptote, which suggests that the sampling effort for all three sites was sufficient.



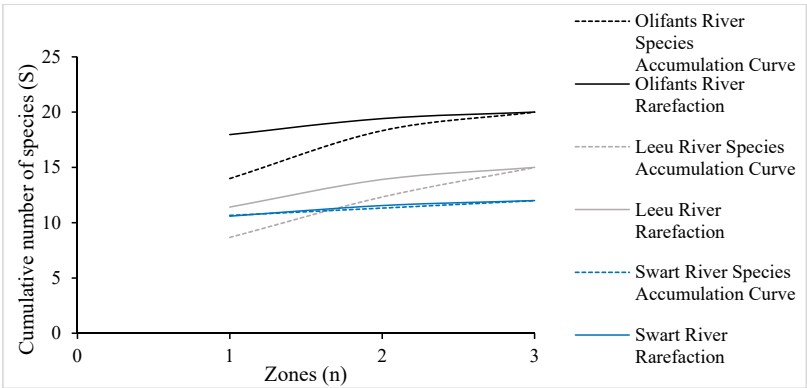

**Figure 1.** Species accumulation (observed, dotted line) and Rarefaction (expected, solid line) curves for riparian plants at Olifants (Black), Leeu (Grey), and Swart (Blue) river, Western Cape, South Africa. The total number of woody species is 20, 15, and 12, respectively. The curves represent successively pooled and randomly ordered samples between three different zones sampled. Curves were computed using Estimate S [33].

*3.3. Species Richness*

The Margalef's index ($D_{mg}$) for species richness showed that the Olifants and Leeu river species richness increased from the waterpoint outwards in contrast to the Swart River species richness, which decreased from the river bank outwards (Figure 2). The species richness indices for all the sites were <3. The Leeu River had the highest species richness index, across all the three zones, compared to the other rivers sampled. Two-way ANOVA tests indicated that there was an overall significant difference ($p < 0.05$) between species richness at the site and zonal level.

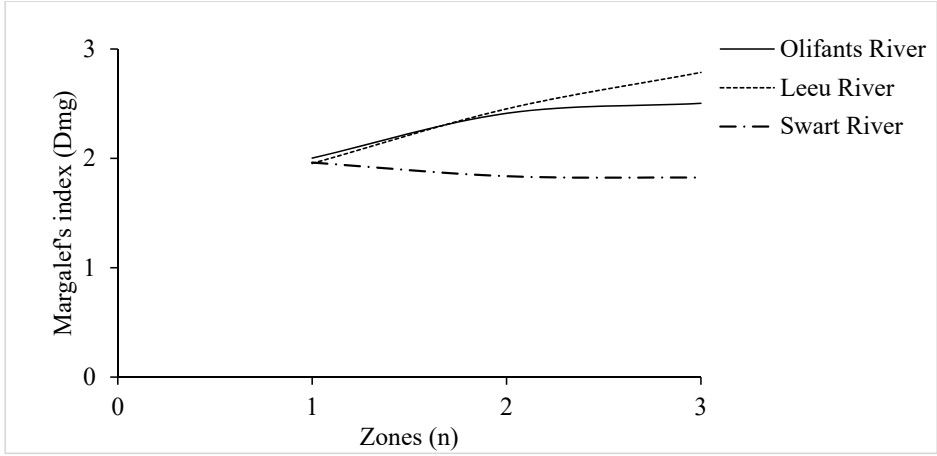

**Figure 2.** The cumulative species richness curves of Margalef's index ($D_{mg}$) for plants at Olifants (solid line), Leeu (dotted line), and Swart (solid-dotted line) rivers, Western Cape, South Africa. The curves represent successively pooled and randomly ordered samples at three zones sampled. Curves were computed using Estimate S [33]. The formula for the index is $D_{mg} = (S - 1)/\ln N$.

*3.4. Species Diversity*

The Shannon diversity index (H′) measures the level of uncertainty in correctly predicting the identity of the next species chosen at random. The Shannon diversity index increased from the waterpoint outwards in all three sites (Figure 3a). The Leeu riparian zones had the highest species richness, i.e., the highest degree of uncertainty in predicting the identity of the next species chosen at random, and, therefore, the highest diversity. The Simpsons diversity index ($-\ln\lambda$) along the Leeu riparian zone increased slightly from the waterpoint outwards, whereas that of the Olifants and Swart

riparian zones showed no increase moving outward from the waterpoint (Figure 3b). The zone furthest away from the waterpoint had the highest species diversity. The Olifants riparian zone had the highest dominance (i.e., the vascular plant community was mostly made up of a single species) compared to the Leeu and Swart riparian zones. High dominance (low −lnλ value) is indicative of lower species diversity (Figure 3b). Hill's Diversity Number (N1), which shows diversity in terms of the effective number of species that are present within a sample indicates that the Leeu riparian zone species diversity is likely to increase away from the river bank than the Olifants and Swart riparian zone species diversity, which are more likely to remain low and unchanged (Figure 3c). These findings corroborate those of the Shannon and Simpsons diversity index, which showed that the Leeu riparian zone had the most diverse population. The Leeu riparian zone showed an increasing Fisher's alpha diversity index ($\alpha$) away from the waterpoint, whereas the $\alpha$ of the Swart riparian zone decreased away from the waterpoint. Furthermore, the Olifants riparian zone $\alpha$ plateaus away from the waterpoint (Figure 3d). Two-way ANOVA tests conducted for the diversity indices indicated that between the three zones and sites species diversity differed significantly ($p < 0.001$). In addition to these findings, the one-way ANOVAs showed that for the Simpson's diversity index, the Olifants River diversity was significantly lower ($p < 0.05$) compared to both the Swart and Leeu river species diversity.

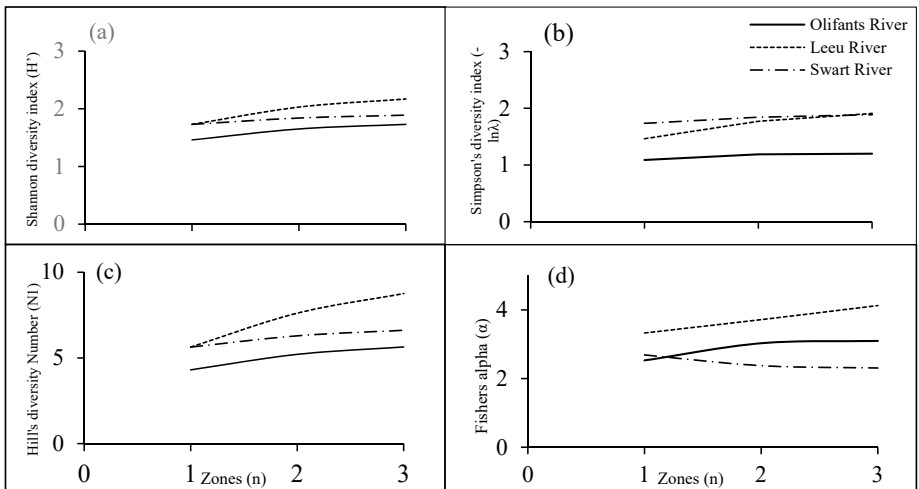

**Figure 3.** Species diversity curves for Olifants (solid line), Leeu (dotted line) and Swart (solid-dotted line) rivers in Western Cape, South Africa. (**a**) The Shannon diversity index curves with the riparian zone has the highest index, (**b**) the Simpson's diversity index curves with the Leeu River and Swart riparian zones plateau at the same highest value (**c**) Hill's diversity number curves show the Leeu riparian zone to have the highest diversity value and (**d**) the Fisher's alpha index is the highest for the Leeu riparian zone with the Swart River Fisher's alpha value decreasing away from the river bank. Species diversity indices increased towards the zone furthest from the waterpoint.

### 3.5. Species Evenness

The Leeu riparian zone had the highest Evenness index than the Olifants and Swart riparian zones (Figure 4), and across all sites, Zone 1 diverged from evenness. These results indicate that the Leeu riparian zone species compositions have the lowest dominance of species. The two-way ANOVA tests showed that, overall, the evenness differed significantly ($p < 0.001$) between zones and sites, the one-way ANOVAs corroborated these findings showing that Olifants River species composition was significantly less even than the Swart and Leeu rivers species composition.

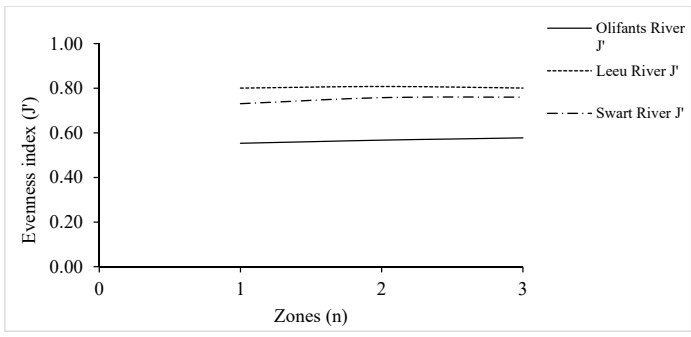

**Figure 4.** Evenness curves for Olifants (solid line), Leeu (dotted line) and Swart (solid-dotted line) rivers in Western Cape, South Africa. Evenness index was the highest along the Leeu riparian zone.

### 3.6. Rank Abundance Curves

The rank abundance curves of the three sites sampled were all steep, which is a pattern indicative of uneven plant communities (Figure 5). Only the top three most abundant plant species are displayed in the curves as follows: (a) along the Swart River: (i) Exotic *Tamarix*, (ii) *Lampranthus uniflorus*, and (iii) *Atriplex muelleri*; (b) along the Olifants River: (i) *Oncosiphon piluliferum*, (ii) Brassicaceae, and (iii) Exotic *Tamarix*; (c) along the Leeu River: (i) *Mesembryanthemum crystallinum*, (ii) *Lycium oxycarpum*, and (iii) *Prosopis juliflora*.

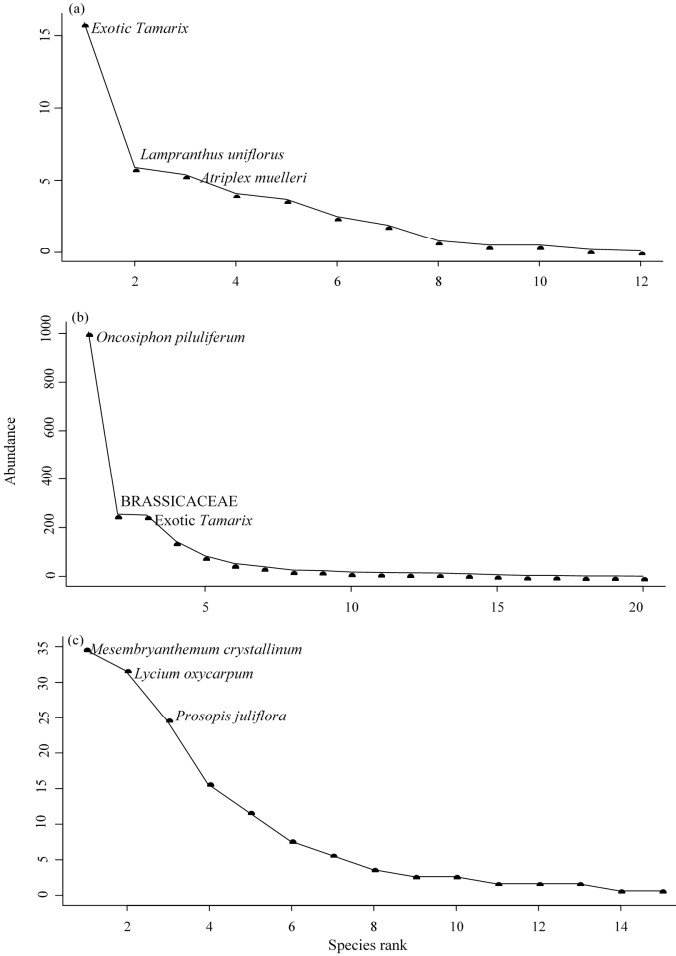

**Figure 5.** Rank abundance curves showing the top three most abundant species for (**a**) Swart, (**b**) Olifants, and (**c**) Leeu rivers, Western Cape, South Africa. The curves are all steep, and exotic *Tamarix* only occurs in the top three most abundant species along the Swart and Olifants rivers.

### 3.7. Complementarity and Similarity

The Jaccards and Sorenson's similarity index between the three sampled rivers were all low ($\bar{x} < 27\%$) (Table 2). This indicates that vascular plant species composition varied greatly between the sites that were invaded by exotic *Tamarix* species. Species composition was less similar between the Leeu River (which had the highest species richness, diversity, and evenness) and the Swart River (which had low species richness, diversity, and evenness). Between the three zones sampled along the Leeu, Olifants, and Swart rivers, the Jaccards and Sorenson's similarity index indicated that the Swart riparian zone plant community had the highest species similarity between the zones ($\bar{x} > 80\%$) (Table 3). The Leeu riparian zone plant community had low species similarity between the zones ($\bar{x} < 50\%$) (Table 3). Similarly, the Olifants riparian zone plant community complementarity and similarity indices were indicative of a community, across all three zones, with low degree of similarity ($\bar{x} < 59\%$) (Table 3). Species similarities in all the three sites were relatively low between the first and the third zones, although along the Leeu River, Zone 1 and 3 had the highest species similarities, whereas along both the Olifants and Swart river species, the similarities were highest between Zone 2 and 3.

**Table 2.** The Jaccard's and Sorenson's coefficients of similarity between the Leeu, Olifants, and Swart riparian zones, expressed as percentages.

| Similarity and Complementarity Index | Leeu River | Swart River |
|:---:|:---:|:---:|
| **Jaccard Index** | | |
| Olifants River | 21 | 23 |
| Leeu River | | 17 |
| **Sorenson's index** | | |
| Olifants River | 34 | 38 |
| Leeu River | | 29 |

**Table 3.** The Jaccard's and Sorenson's coefficients of similarity between the three zones sampled along the Leeu, Olifants, and Swart rivers, expressed as percentages.

| | Leeu River | | Olifants River | | Swart River | |
|:---:|:---:|:---:|:---:|:---:|:---:|:---:|
| Similarity and Complementarity Index | Zone 2 | Zone 3 | Zone 2 | Zone 3 | Zone 2 | Zone 3 |
| **Jaccard Index** | | | | | | |
| Zone 1 | 36 | 50 | 44 | 41 | 83 | 83 |
| Zone 2 | - | 36 | - | 70 | - | 100 |
| **Sorenson's index** | | | | | | |
| Zone 1 | 53 | 67 | 62 | 52 | 91 | 91 |
| Zone 2 | - | 53 | - | 82 | - | 100 |

## 4. Discussion

Invasive alien plants are strong drivers of biodiversity change or loss, and exotic *Tamarix* species are such plants, with the ability to modify ecosystems, and this has become evident through research conducted in the Western Cape Province. The Western Cape Province is known to have the highest exotic species (*T. chinensis*—Tc and *T. ramossissima*—Tr) and exotic hybrid species (Tc × Tr) populations in comparison to other South African provinces [36]. This finding is alarming in that Mayonde et al. [22] found hybrid *Tamarix* species (Tc × Tr) to be the dominant (64.7%) invasive genotype. Hybridisation is an evolutionary trait that invasive organisms have developed in order to rapidly increase genetic variation and possibly invasion vigour as well.

Complementarity and similarity assessments of species between the study sites indicated that the Leeu River was highly dissimilar to both the Olifants and Swart rivers with both the Jaccard's and Sorenson's coefficients of similarity being the lowest for the Leeu River (Table 2). A closer look at each individual site showed that the Swart River and the Olifants River had higher species similarity between zones than the Leeu River, which had the lowest zonal species similarity (Table 3).

High complementarity and similarity between zones are suggestive of a riparian community that is made up of a high occurrence of a similar species. The species accumulation and rarefaction curves showed that the Swart River had a homogenous riparian zone plant community (Figure 1). In both, the case of the Olifants River and the Swart River, exotic *Tamarix* was identified as the most abundant tree species. This finding was supported by the rank abundance curves where exotic *Tamarix* ranked first and third along the Swart and Olifants River, respectively (Figure 5).

Newete et al. [13] conducted a study to establish the distribution and abundance of the invasive *Tamarix* genotypes in South Africa and found a strong negative relationship (linear correlation) between the density of the exotic *Tamarix* species and co-occurring tree and shrub species. The Olifants and Swart rivers had the highest exotic *Tamarix* density across the 11 riparian zones sampled [13]. This study developed on these findings to establish the knowledge gaps on how biodiversity, with regards to species richness, diversity, and evenness, was impacted, and which parts of the riparian zone were most vulnerable to invasion by exotic *Tamarix*. It became evident in this study that the Swart and Olifants rivers had lower species richness, diversity, and evenness than the Leeu River, which consistently had the highest biodiversity indices (Figures 2–4). The findings observed at the Swart and Olifants riparian zones were similar to the trends observed in studies conducted in the United States of America where biodiversity decreased in areas where invasion by exotic *Tamarix* increased [26,37].

This study further highlights that Zone 1 across all the three sites had the lowest biodiversity. This is indicated by the gradual increase and peak, at Zone 3, for all biodiversity indices, except for the Fishers alpha index at the Swart River, which decreased from the waterpoint outward (Figures 2–4). The low biodiversity in Zone 1 is an indication of a high abundance of exotic *Tamarix* near the water source. These findings allude to the notion that exotic *Tamarix* is well adapted to displace native plants that naturally take advantage of the phreatic zone as an immediate water source. *Tamarix* shrubs and trees have roots that can extend into alluvial deposits, allowing them access into the groundwater; this further allows them to outcompete native shrubs and trees for water [38–40]. In addition, they form dense stands which are not only less penetrable to animals, but they outcompete native flora for light, this is similar to the observations made by Steenkamp and Chown [38] as well as van Klinken et al. [39] regarding the invasive *Prosopis* species.

The higher abundance of exotic *Tamarix* trees in Zone 1 can most likely result in narrower river banks which can alternatively contribute to the drying out of rivers; such consequences are even more detrimental for areas where the affected rivers are a direct source of water, e.g., the Leeu River feeds into the Gamka Dam, which provides water for the Beaufort West community of >51,000 people and their animals [40]. *Tamarix* species also shed leaves high in salt content, as they are able to accumulate and transport salt from the soil into their leaves [13,26]. The leaf litter acts as fire fuel or excrete salts that remain on the soil surface, inhibiting the germination and growth of native competing flora [41]. In addition, exotic *Tamarix* trees becoming denser towards the water source compete with trees whose growth relies on the catchment, this is a typical characteristic of *V. karroo*, *S. pendulina*, *T. usneoides*, *D. lycioides*, and *L. oxycarpum*, which were sampled at the various sites. We can, thus conclude that exotic *Tamarix* species are vigorous invaders and can act as modifying species in the ecosystems they occur in.

The Leeu River was the only site sampled where the native congeneric to the exotic *Tamarix* species, *Tamarix usneoides*, grew. *Tamarix usneoides* could be acting as a direct competitor to its exotic counterparts because they grow to above-ground heights and root lengths that are similar, and, therefore, they use the same water sources. Native congeneric species are also well adapted to the conditions imposed by their exotic counterparts, e.g., *T. usneoides* is a better salt accumulator than exotic *Tamarix* and their hybrids [36], and, therefore, an increase in soil salt concentrations through invasion does not impact the abundance of *T. usneoides*. It was also the only site were exotic *Tamarix* did not rank among the top three most abundant species. Instead, *Prosopis juliflora*, ranked as the third most abundant tree at the riparian zone. These findings suggest that invasion by exotic *Tamarix* species at a site where they are exposed to other competitors, ideally phreatophytes, halophytes, or xerophytes, have compromised competitive

ability. Along the Leeu riparian zone, *Mesembryanthemum crystallinum* and *Lycium oxycarpum* ranked first and second, respectively. This was the only site where native shrub species were more abundant than exotic *Tamarix* trees (Figure 5).

The Swart riparian zone was not only invaded by exotic *Tamarix*, which ranked as the most abundant vascular plant but *Atriplex muelleri* an exotic shrub native to Australia, ranked as the third most abundant plant species (Figure 5a). This finding suggests that at riparian zones where exotic *Tamarix* is the most abundant species, other alien and potentially invasive woody species are most likely to establish. This is even more likely in arid areas like the Little Karroo, which are vulnerable to invasion by exotic plant species [42]. *Lampranthus uniflorus* (succulent shrub) was the only native and perennial species among the top three most abundant species. *Lampranthus uniflorus* is an indicator species for highly degraded habitats and arid environmental conditions [29]. The shrub also provides evidence for impacts of environmental change in an area, giving more information about the biotic or abiotic state of the environment and predicting how diverse other plants species, taxa and communities, are in an area. The high abundance of *Lampranthus uniflorus* is most likely influenced by changed soil characteristics that are a result of exotic *Tamarix* invasion, which is known to increase soil salt levels [43]. This plant is not only alerting of potentially irreparable damage at this site, but it provides a red flag for an urgent need of biodiversity conservation in terms of species richness, diversity, and evenness as well as land rehabilitation.

The majority of the plants that were sampled where exotic *Tamarix* was the most abundant species were indigenous to South Africa and common in the Little Karroo, Western Cape. A matter of concern is that at these sites, the native vascular flora that ranks among the top three most abundant species, particularly along the Olifants riparian zone (Figure 5b), are classified as herbs (Table 1). Both *Oncosiphon piluliferum* and a species from the Brassiaceae have annual life cycles, and, therefore, are at risk of extirpation should natural disasters (e.g., floods and severe droughts) affect environmental and ecological dynamics that allow for the native plants to thrive along the riparian zone. They are both most likely only planted as ornamentals, and, therefore, contribute very little towards the ecosystem functions and services that occur along the river.

Three grass species were identified at the Leeu River; *Cenchrus* sp. (Subfamily: Panicoideae) is a flammable grass genus, and species in the genus are well adapted to dry area conditions and fire alike [44,45]. *Cynodon dactylon* (Subfamily: Cynodonteae), on the other hand, grows well and its seedlings can re-establish in areas prone to seasonal flooding as observed in the Fafan valley of the Jijinga rangelands, Ethiopia [46]. *Stipagrostis namaquensis* (Subfamily: Aristideae) is a typical Nama Karoo riparian grass, which is able to survive maximum disturbance with unpredictable flooding episodes [47].

Grasses adapted to floods, fire, warm, and dry weather conditions dominate areas that have been invaded by exotic *Tamarix*. Along the Olifants River, in addition to the three species described above, *Enneapogon desvauxii* (Subfamily: Pappophoreae) and *Hordeum murinum* (Subfamily: Triticeae) were also sampled. Both grow well in arid environments that were highly degraded [46,48]. However, should disturbance continue for long periods with increasing severity, the grass populations are prone to decline rapidly. This will have direct implications on biodiversity loss, and, therefore, the loss of ecosystem functions and services provided by the grass populations in the study sites.

Indigenous species that were sampled across the three sites were mostly small succulent shrubs or herbs with, generally, lower water consumption than the exotic *Tamarix*. The large native trees, which occurred more often in Zone 2 and Zone 3 of the Leeu riparian zone (i.e., *T. usneoides*, *L. oxycarpum* and *L. hirsutum*) most likely had resource acquisition strategies, which allowed them to compete with exotic *Tamarix*. These strategies could range from occupying a different niche from the exotic *Tamarix* or capitalising on their tolerance to arid and saline conditions through morphological, physiological, or biochemical adaptations.

Morphological adaptions include having deep roots that allow the plants to penetrate deep water sources or having thorns that limit the extent of herbivory, which, if extensive, could result

in plant population decline [49,50]. Physiological adaptations include modifications of plants to their $C_4/C_3$ metabolism pathways of carbon fixation or leaf size and water use efficiency [51,52]. Biochemical adaptations are facilitated by multiple biochemical pathways that facilitate water retention and acquisition, while also protecting chloroplast functions and maintaining ion homeostasis [53].

Unlike the Succulent Karoo, which boasts a wide variety of succulent flora that provide social benefits to the surrounding communities [54], the Little Karoo flora leans more towards provisioning and regulating ecosystem services [55]. This is evident in the host of indigenous trees and shrubs that grew along the riparian areas of the latter region, e.g., *Diospyros lycioides* is a popular forage plant, especially in the Karoo, where forage is scarce, particularly during the dry seasons or extreme drought periods, which the plant is well adapted to [56].

The economic and ecological resources that are provided by the various plants that grew along the sampled riparian zones vary greatly between the different plants (Table 4). Some of the economic resources include grazing, medicinal value, aromatic source, edible food (for humans or animals), fuelwood, fencing, ornamental value, making handicrafts, providing shelter, rope making, and thatching. Ecological resources include sand stabilization, refuge provision, shading, soil fertility, salt tolerance, windbreak, waterpoint stabilisation, and other resources such as bioremediation and pollen supply [57].

**Table 4.** Summary of the economic and ecological resources provided by some of the trees, shrubs, and herbs sampled along the Swart, Olifants, and Leeu riparian zones [57]. Economic sources include: Grazing (Gr), Medicinal (Md), Aromatic source (Ar), Edible food (Ed), Fuelwood (Fu), Fencing+ Windbreak (Fe), Ornamental (Or), Making handicrafts (Han), and other uses, such as shelter, rope making, and thatching (Ot). Ecological resources include Sand stabilization (Sf), Refuge (Re), Shading (Sh), Soil fertility (Sr), Salt tolerance (St), Wind break (Wb), Weed (We), Waterpoint stabilisation (Rs), and other resources such as bioremediation and pollen supply (Ot).

| Plant | Economic Resources | Ecological Resources |
|---|---|---|
| *Atriplex semibaccata* | Gr, Md, Ed | Sf, We |
| *Chenopodium murale* | Md, Ed, Ar | Sf |
| *Cynodon dactylon* | Gr, Md, Fu | Sf, Re, Sh |
| *Hordeum murinum* | Gr, Md | Sf |
| *Juncus kraussii* | Gr, Han, Ot | Sf, Re, St, Wb, Ot |
| *Lycium hirsutum* | Gr, Md, Ed, Fu, Fe, Ot | Sf, Re, Sh, Sr, Wb, Rs |
| *Lycium oxycarpum* | Gr, Md, Ed, Fu, Fe, Ot | Sf, Re, Sh, Sr, Wb, Rs |
| *Medicago sativa* | Gr | Sr |
| *Salsola barbata* | Gr, Md, Ed, Fu | Ot |
| *Senecio burchellii* | Gr, Md, Ed | Sf, We |
| *Sisymbrium* sp. | Gr | - |
| *Solanum tomentosum* | Md | Sf |
| *Stipagrostis namaquensis* | Gr, Md | Sf, Sr |
| *Tamarix usneoides* | Gr, Md, Fu, Or, Ot | Sf, Re, Sh, Wb, Rs |
| *Zygophyllum retrofractum* | Md, Ot | Sf, St, Wb, Rs |

Given the various uses of the flora growing across the riparian zones, losing elements of biodiversity as a result of displacement by the invasion of exotic *Tamarix* species negatively influences the maintenance and resilience of invaded riparian zones. Several plants have been identified as key sources of pollen. The Little Karroo does not boast a wide range of flowering plants, and, therefore, pollen sources are usually scarce yet valuable [55]. This has direct implications on industries that rely on pollinators, such as agriculture and the textile industry. Pollination is a vital process for seed and fruit production and the overall plant life cycle [58]. The site sampled along the Olifants River is adjacent to a farm. Studies conducted by Kremen et al. [59] and Klein et al. [58] showed that farms next to natural habitats attracted more pollinators, and therefore, resulted in higher crop yield. For areas like the Little and Central Karoo, where an arable land is a form of subsistence and gaining revenue,

this will have negative impacts on the livelihood of inhabitants in these communities. A loss in biodiversity will have direct implications for food security.

Rutherford and Powrie [29] suggested that severe degradation of land, due to heavy grazing, does, in fact, result in the decrease of species but can ultimately increase the diversity and evenness of indigenous plant species. This was especially true for sites where land was vacant and accessible to plant species such as *Oncosiphon piluliferum*, *Drosanthemum framesii*, and *Galenia sarcophylla*, which are well adapted to overgrazed land [29]. This observation would not be true for sites that have been degraded by IAPs, like exotic *Tamarix*, as they reduce biodiversity by occupying and modifying habitats and ecosystems, making them inhabitable for the bulk of indigenous species. This outcome has become evident in this study, particularly along the Swart and Olifants riparian zones, where biodiversity at both sites decreased and was lower in comparison to the Leeu riparian zone.

Prioritising the rapid removal of exotic *Tamarix* species will gradually allow for the restoration of ecosystem services and functions that are being rapidly lost as the exotic *Tamarix* extends across and along the riparian zones. Not only will ecosystem disturbances become more frequent, it will become easier for other exotic species to establish as they easily take advantage of the degraded and disturbed sites. This is common of smaller herbs and weed plants (e.g., *Argemone* sp.) and other larger woody plant species, which have commonly established easily in degraded sites (e.g., *Prosopis* sp.) [60].

Dwarf succulent shrubs are characteristic of and endemic to the Little and Central Karoo woody plant flora [55]. Indigenous larger trees introduced into riparian zones to stabilise waterpoints amongst other hosts of functions do not typically encroach ranges beyond their introduced ranges. The limited range of the native larger trees grants exotic *Tamarix* species an opportunity to expand into ranges beyond the riparian zone as they are not limited to just the riparian zones; this is due to the plants not being obligate phreatophytes [61,62]. In the long-term, other ecosystems (e.g., wetlands) will most likely become more vulnerable to invasion, and the extirpation, extinction, or a marked local decline of plant populations of other indigenous and endemic vascular plant species will become even more evident.

## 5. Conclusions

The introduction of exotic *Tamarix* species into South Africa has caused an imbalance in normal ecosystem functions and services. This study further highlighted the importance of thorough consideration of plants selected for phytoremediation purposes. Plants that are not native to novel sites pose a higher threat to biodiversity, while native (hyper) accumulators do not affect ecosystem functions and services negatively. Studies have shown that native congeneric species (e.g., *T. usneoides*) are usually better options for phytoremediation practises, and, therefore, their use should always be prioritised. *Vachellia karroo*, *Lycium oxycarpum*, and *Lampranthus uniflorus*, *Cenchrus* sp., *Cynodon dactylon*, *Stipagrostis namaquensis*, *Enneapogon desvauxii*, and *Hordeum murinum* have been identified as native vascular plants that can be re-planted on previously invaded land since they have been identified as resilient and suited to saline and arid environmental conditions while providing essential roles to the environment.

Successful revegetation projects can only be achieved with constant and consistent post-removal assessments, ensuring that native vascular plant species are establishing and performing the identified ecosystem services and functions. It is the role and responsibility of policy-advisers and policymakers to take into consideration research outputs that serve to conserve biodiversity, and it is the collective role of society at large to call out practises that result in biodiversity loss, especially plant biodiversity.

**Author Contributions:** Conceptualization, K.T.A.S. and S.W.N.; Data curation, K.T.A.S. and S.W.N.; Formal analysis, K.T.A.S.; Funding acquisition, S.W.N.; Investigation, K.T.A.S.; Methodology, K.T.A.S. and S.W.N.; Supervision, S.W.N.; Writing—original draft, K.T.A.S.; Writing—review & editing, K.T.A.S. and S.W.N. All authors have read and agreed to the published version of the manuscript.

**Funding:** This research was funded by National Research Foundation (NRF), grant number 114345.

**Acknowledgments:** The Agricultural Research Council (ARC) and the Postgraduate Merit Award (PMA) at the University of the Witwatersrand for the financial support as well as Prof. Sue Milton for plant identification.

**Conflicts of Interest:** The authors declare no conflict of interest. The funders had no role in the design of the study; in the collection, analyses, or interpretation of data; in the writing of the manuscript, or in the decision to publish the results.

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
