# Peer review of "The Impact of Exotic Tamarix Species on Riparian Plant Biodiversity"

_agriculture, doi:10.3390/agriculture10090395_

Round 1
Reviewer 1 Report
The manuscript is well presented and worthy of publication, but after correcting the required revisions. Furthermore, it is appropriate to specify some concepts required in the text, for example,1) differences between invasive and alien species, 2) specify if there are differences between the invasive species of Tamarix detected, 3) vegetational surveys would have been appropriate (Braun_Blanquet 1932 method) to better understand some dynamics of the impact on river vegetation. However, this last consideration is personal and not have any negative effect about the acceptance of the manuscript that the author presented.
in the attached file highlighted in yellow color the periods commented that require attention from the authors.

Author Response
The manuscript is well presented and worthy of publication, but after correcting the required revisions. Furthermore, it is appropriate to specify some concepts required in the text, for
example,1) differences between invasive and alien species,
Response: added – (not endemic or indigenous to South Africa) line 163
Alien species in this context refers to those that are not endemic or indigenous to South Africa or the country with which reference was made to whereas invasive species are those that have been classified under the invasive species lists from the National Environmental Management: Biodiversity Act (NEM: BA) (Act No. 10 of 2004) or Conservation of Agricultural Resources Act (CARA) (Act No. 43 of 1983) legislature
2) specify if there are differences between the invasive species of Tamarix detected
Response: Invasive species of Tamarix here are referred to as a collective because they were not flowering and thus could only be distinguished using the bark/stem colour which is different only to the indigenous Tamarix species. Tamarix chinensis and T. ramossissima are known to co-occur particularly in the Western Cape. Mayonde et al. (2016) conducted a study that used genetic analysis to identify specific Tamarix species.
3) vegetational surveys would have been
appropriate (Braun_Blanquet 1932 method) to better understand some dynamics of the impact on river vegetation. However, this last consideration is personal and not have any
negative effect about the acceptance of the manuscript that the author presented. in the attached file highlighted in yellow color the periods commented that require attention from the authors.
Response: noted
Biodiversity It’s better the word communities, because the the concept of biodiversity it’s different. So, I suggest to change the term biodiversity with communities
Response: Measurements of biodiversity often include: (i) richness – also known as species composition, it refers to the number of unique species that occur in an area, (ii) evenness – measures the equitability among different species and (iii) heterogeneity – assesses the dissimilarity among species that exist in an area (Cardinale et al., 2012). The term biodiversity was used based on this premise and that analysis used in this study specifically focused on the 3 mentioned measures of biodiversity
it is preferable to synthesize
Response: Much of the results have already been summed up to the main findings represented here
Please ad 2 reference: i.e. Viciani D., Vidali M., Gigante D., Bolpagni R., Villani M.C., Acosta A.T.R., Adorni M., Aleffi M., Allegrezza M., Angiolini C., Assini S., Bagella S,. Bonari G., Bovio M., Bracco F., Brundu G., Buffa G., Caccianiga M., Carnevali L., Ceschin S., Ciaschetti G., Cogoni A., Di Cecco V., Foggi B., Frattaroli A.R., Genovesi P., Gentili R., Lazzaro L., Lonati M., Lucchese F., Mainetti A., Mariotti M., Minissale P., Paura B., Pellizzari M., Perrino E.V., Pirone G., Poggio L., Poldini L., Poponessi S., Prisco I., Prosser F., Puglisi M., Rosati L., Selvaggi A., Sottovia L., Spampinato G., Stanisci A., Stinca A., Venanzoni R., Lastrucci L., 2020. A first checklist of the alien-dominated vegetation in Italy. Plant Sociology 57(1): 29–54. DOI: 10.3897/pls2020571/04
Response: added (Pamela, S.E., 2000. The root causes of biodiversity loss. Earthscan.) and (Viciani D., Vidali M., Gigante D., Bolpagni R., Villani M.C., Acosta A.T.R., Adorni M., Aleffi M., Allegrezza M., Angiolini C., Assini S., Bagella S,. Bonari G., Bovio M., Bracco F., Brundu G., Buffa G., Caccianiga M., Carnevali L., Ceschin S., Ciaschetti G., Cogoni A., Di Cecco V., Foggi B., Frattaroli A.R., Genovesi P., Gentili R., Lazzaro L., Lonati M., Lucchese F., Mainetti A., Mariotti M., Minissale P., Paura B., Pellizzari M., Perrino E.V., Pirone G., Poggio L., Poldini L., Poponessi S., Prisco I., Prosser F., Puglisi M., Rosati L., Selvaggi A., Sottovia L., Spampinato G., Stanisci A., Stinca A., Venanzoni R., Lastrucci L., 2020. A first checklist of the alien-dominated vegetation in Italy. Plant Sociology 57(1): 29–54. DOI: 10.3897/pls2020571/04))
America Sure? I have some doubt
Response: Hultine et al. (2015) confirms this statement
Please check the period about size and font Please format the period about size and font
Response: It is not clear what the reviewer has asked for here line 73
It’s ok, but there are some national initiative or project about it? If yes please add two words
Response: Newete et al. (2019a) conducted a study that compared the density of exotic Tamarix species to co-occurring plant however, this study did not distinguish between native and alien or invasive plant species.
it would be better to insert a location map. …For a reader who does not know south africa. what do you think about it?
Response: Location map was left out of this manuscript to emphasise only graphics that appear I the results section
Please specify what the authors want to means. Geographic zones or habitat zones? In this way it's not clear for the readers. Maybe rivers zones?
Response: mentioned in line 125-126 under methods section
Added “Zones where classified as distances across the river bank with similar species composition i.e. (Zone 1: 0-15m; Zone 2:15-35 and Zone 3: >35m)” line 181-183 for reiteration
The name of the authors for each species when you cited for the first time it in the text. Remember previous comment.
Response: Name of authors cited in Table 1
Some data of all exotic tamarix in swart river?
Response: added “showing the top tree most abundant species”. not clear what the comment is asking.
In the all text the authors cited always exotic Tamarix, without differences into species of Tamarix. There are no differences ability to modify ecosystems for each Tamarix species that you considered in your study? If yes tow words about it
Response: the impact of invasion by exotic Tamarix species is not different amongst the various species found in the country however the density of the hybrid exotic species is higher that the other Tamarix subtypes that occur here.
Added “ The Western Cape Province is known to have the highest exotic species (T. chinensis - Tc and T. ramossissima - Tr) and exotic hybrid species (Tc x Tr) populations in comparison to other South African provinces (Newete et al, 2019b). This finding is alarming in that Mayonde et al. (2015) found hybrid Tamarix species (Tc x Tr) to be the dominant (64.7%) invasive genotype. Hybridization is an evolutionary trait that invasive organisms have developed in order to rapidly increase genetic variation and possibly invasion vigour as well.” Line 279-284
Plant community, but which one? Please at least it’s welcome a syntaxonomic reference in the absence of phytosociological samples
Response:

Reviewer 2 Report
Setshedi and Newete present an interesting study of the effects of Tamarix invasions on the riparian vegetation in three rivers in South Africa. This is generally a nice and informative manuscript, which I would like to see eventually published.
Nevertheless, I find many serious faults in presentation and writing. Apart from some language errors (mainly involving typing and punctuation), the quality of writing is insufficient. There are some inconsistencies in the results section, which must be resolved.
General comments:
- Please use line numbering, unless clearly instructed not to. It is difficult to track and point comments with no line numbers.
- The authors should complete the Author Contribution, Funding and Conflict of Interest statements.
Title:
- “Tamarix the triumphant” does not really say anything, and even implies that this “triumph” is a good thing. I would change the title to be more informative and accurate, something in the lines of “Invasive Tamarix species (do this and that) in riparian ecosystems”.
Abstract:
- Please indicate shortly what is the difference between Zones 2 and 3.
- It is difficult to follow the river names. What are the principal differences between the sites (invaded vs control, Climate, Invasion intensity, Dominant species, etc.)?
- You mention displacement of herbaceous species, but earlier write you only surveyed woody plants. And then at the very last sentence pollinators join in. It looks speculative.
Introduction:
- Paragraph 1: This could be a much shorter paragraph focusing on effects of biodiversity loss on ecosystem services. There is no real need to mention over-population.
- Paragraph 2: “transformation of natural areas” is too general. Do you include invasions in this? I would not.
- Do not mix exotic and invasive. Use one term, or else clearly define what is the difference between the two.
- Although I appreciate the style in “inter alia”, I would avoid overusing this term for the benefit of those who are less fluent in Latin.
- “Invasive alien plants … flooding” requires further elaboration on the mechanism, and a reference. Although an explanation follows (not explicitly referring to floods), this sentence feels detached.
- I would place paragraphs 3-5 before paragraph 2.
- Paragraph 4: The text does not flow too well.
- Paragraph 6: Although important, this paragraph is not sufficiently good. It misses many details (what traits?), and I am not sure about its placing.
- Paragraph 7: This paragraph will require some modifications following other revisions. I would also include ecosystem services as a secondary aim, although carefully since it is not measured directly.
Methods:
- There is no need for headings 2.2.1 if there is no heading 2.2.2
- Study sites: Are there no controls? How do we know what the native vegetation is like?
- Experimental design: What were the transect lengths? How many transects per site? Are there principal differences between zones, or are they determined strictly by distance from water?
- Data analysis: “The Margalef…” – which index is used for what?
- After reading the results, I understand that the authors used distances from the water (zones) as indicators of influence – the farther from the water you are, the more “native” and uninfluenced by the invasion is the vegetation. That means that zone 3 is an internal control per river. If this is the case, this must be spelt out in the methods section.
Results:
- Capitalize only the first letter in family names.
- You note that you surveyed woody plants, but then report 6 grass species. Please clarify.
- Most of the report of species numbers per family is redundant, having Table 1.
- Table 1: Add for each species at which site/s it was found.
- I am unsatisfied by the rarefaction analysis. The x-axis needs to be quantitative and continuous (as much as possible), especially if there are differences in area size among zones and/or sites.
- Figure 1: Describing which curve relates to which case in both a textual and a graphical legend is redundant. Let alone that one river is not shown in the graphical legend. Also, the numbers do not match the text (20, 15 and 12 compared to 13, 25 and 18).
- The authors sometimes refer to Dmg as “species richness index”, which is confusing and misleading.
- Section 3.3: You should show, possibly in a table, which differences are significant and which are not.
- Figures 3+4: It is important to indicate significant differences within the graphs.
- Section 3.6: I do not understand the rationale behind this analysis. Should it not be at the zone scale? What information does it provide?
- Tables 2+3: Why not have two smaller tables (one per index), directly comparing all three rivers or all three zones?
Discussion:
- The first paragraph/sentence is redundant.
- I advise the authors to also read the following:
Katz O, Stavi I. (in press). Hierarchical effects of Tamarix aphylla afforestation in a sand dune environment on vegetation structure and plant diversity. Forest Science, https://doi.org/10.1093/forsci/fxaa011 - Page 11, second paragraph: This is where an external control (Tamarix-free river) would have been very helpful. As it sits now, it cannot be 100% sure that the plant diversity decrease in zone 1 is entirely related to the invasion.
- Page 11, last paragraph + page 12, first paragraph: The occurrences of T. usneiodes and A. muelleri could have been provided in the methods section, instead of misleading readers that all rivers are equal, and then surprising us.
- “This finding suggests that … most likely to establish”: Do you know that Tamarix was the first to establish and the others followed? Is it possible that this river was just more susceptible to invasions?
- The following part about climate change is farfetched. At the very list, give evidence/reference for such a change, preferably show its impact on the studied ecosystems.
- Page 12 onwards: Most of this is too speculative, or at the very least not strongly connected to the main story and evidence and not leading anywhere clearly. The section referring to ecosystem services is important, but not well established. I very much liked the idea of having one, but disliked this particular one.
Author Response
Reviewer # 2
General comments:
Please use line numbering, unless clearly instructed not to. It is difficult to track and point comments with no line numbers. The authors should complete the Author Contribution,
Funding and Conflict of Interest statements.
Response: Author Contributions
Conceptualization, Kgalalelo Setshedi and Solomon Newete; Data curation, Kgalalelo Setshedi and Solomon Newete; Formal analysis, Kgalalelo Setshedi; Funding acquisition, Solomon Newete; Investigation, Kgalalelo Setshedi; Methodology, Kgalalelo Setshedi and Solomon Newete; Supervision, Solomon Newete; Writing – original draft, Kgalalelo Setshedi; Writing – review & editing, Kgalalelo Setshedi and Solomon Newete.
Funding
National Research Foundation (NRF) received through the grant-holder linked student support of the NRF funded research project (Grant No, 114345) under Dr Solomon Newete
No conflict of interest
Title:
“Tamarix the triumphant” does not really say anything, and even implies that this “triumph” is a good thing. I would change the title to be more informative and accurate, something in the lines of “Invasive Tamarix species (do this and that) in riparian ecosystems”.
Response: Title changed to “Impact of Invasive Tamarix on Riparian Plant Biodiversity”
Abstract:
Please indicate shortly what is the difference between Zones 2 and 3. It is difficult to follow the river names.
Response: added “Each transect was split into three zones (Zone 1: 0-15m; Zone 2:15-35 and Zone 3: >35m), which were identified at each site based on species composition across each riparian zone.” Line14-16
What are the principal differences between the sites (invaded vs control, Climate, Invasion intensity, Dominant species, etc.)?
Response: Invasion intensity and a knowledge of Tamarix species that occurred there. Added “which had varying invasion intensities” line 14. Leeu River only had the native Tamarix usneoides.
You mention displacement of herbaceous species, but earlier write you only surveyed woody plants. And then at the very last sentence pollinators join in. It looks speculative.
Response: added “Herbaceous aerial cover (HAC) was determined subjectively and objectified using the Walker aerial cover scale.” Line 17-18
The discussion and conclusion elaborate on the links that occur here.
Introduction:
Paragraph 1: This could be a much shorter paragraph focusing on effects of biodiversity loss on ecosystem services. There is no real need to mention over-population
Response: mention of other factors affecting biodiversity is don’t to notify the reader that biodiversity loss is a consequence of all these factors either acting together or solely.
Paragraph 2: “transformation of natural areas” is too general. Do you include invasions in this? I would not.
Response: changed to “The lack of plant conservation directly cascades into the loss of fauna and entomofauna likewise (Giam et al., 2010).
Do not mix exotic and invasive. Use one term, or else clearly define what is the difference between the two.
Response: exotic means alien and not native to a particular region whereas Invasive means the plant has been listed as an invasive species under legislature governing plant conservation in the region mentioned.
Although I appreciate the style in “inter alia”, I would avoid overusing this term for the benefit of those who are less fluent / in Latin.
Response: used only thrice in the whole text, twice in the abstract to avoid making it too wordy and once in the introduction
“Invasive alien plants … flooding” requires further elaboration on the mechanism, and a reference.
Response: “, and ultimately lead to exposure to flooding.” Removed.
Although an explanation follows (not explicitly referring to floods), this sentence feels detached.
I would place paragraphs 3-5 before paragraph 2.
Response: sentence changed to “Nutrient cycling regimes change as a result of invasion by alien plants, furthermore, ground cover which provides surface stability is lost this eventually leads to an increase in soil erosion (Witkowski, 1991; Holmes et al., 2005).”
Paragraph 4: The text does not flow too well.
Response: line 70-82, added Tamarix authors
Over 10 million ha of land in South Africa is invaded by alien plants (van Wilgen et al., 2001) and Tamarix L. , is one of them. Although, the country has an indigenous Tamarix species (T. usneoides), the two exotic Tamarix species (Tamarix ramosissima L and T. chinensis L.) have been in the country close to a century after their first introduction reportedly as garden plants for ornamental purposes (Marlin et al., 2017;). The two species along with their hybrids are currently threatening many of the riparian ecosystem in the country and they are listed under the National Environmental Management: Biodiversity Act 2014 (NEM: BA) as category 1b invasive weed requiring urgent management intervention (Newete et al., 2020). (Newete et al., 2019a). Alien Tamarix species are also considered as one of the 100 worst invasive species globally, under the International Union for Conservation of Nature (IUCN) list of Global Invasive Species Database (Lowe et al., 2000; GISD, 2018).
Paragraph 6: Although important, this paragraph is not sufficiently good. It misses many details (what traits?), and I am not sure about its placing.
Response: Pyšek and Richardson (2008) identified key traits associated with the invasivability of many alien plants that give them a competitive advantage over indigenous plants species by comparing 64 IAPs. Downey and Richardson (2016) defined six thresholds along the extinction trajectory and found that, although no plants are extinct in the wild or globally due to alien plant invasions, native plants have crossed other thresholds along the extinction trajectory due to alien plant
invasions. The decline in native plant populations and extirpations due to exotic Tamarix invasion in South Africa is yet to be quantified. Newete et al. (2019a) compared the density of exotic Tamarix species to the co-occurring plants and found Tamarix density was greater in 7 of the 11 sites investigated.
Moved to line 59
Paragraph 7: This paragraph will require some modifications following other revisions. I would also include ecosystem services as a secondary aim, although carefully since it is not measured directly.
Response:
Methods:
There is no need for headings 2.2.1 if there is no heading 2.2.2
Response: Removed 2.1.1
Study sites: Are there no controls? How do we know what the native vegetation is like?
Response: this study was part of an MSc dissertation. while, a control would have been absolutely important to compare the results with, unfortunately, it was not considered in the survey. partly because the issue was only raised later in the study after the data collection started, and partly because the river banks where the study was conducted were invaded extensively, and we could not find a convenient spot without Tamarix, to select 'a control site' along the same river in a reasonable distance from the treatment.
Experimental design: What were the transect lengths? How many transects per site? Are there principal differences between zones, or are they determined strictly by distance from water?
Response: transects varied in numbers: 3 – 6 belt transect, length: 50m – 75m and width: 25m – 50m line 127, the range has been provided
Data analysis: “The Margalef…” – which index is used for what?
Response: the name of the indices states what it is used for, i.e. The Margalef index (Dmg) for species richness, Shannon-Weiner diversity index (H’), the Simpson diversity index (-lnλ), Hills diversity number (H1), Fisher’s alpha diversity index (α) and the Shannon evenness index (J’)
After reading the results, I understand that the authors used distances from the water (zones) as indicators of influence –
the farther from the water you are, the more “native” and uninfluenced by the invasion is the vegetation. That means that zone 3 is an internal control per river. If this is the case, this must be spelt out in the methods section.
Response: the further away from the river bank meant the lower the density of Tamarix species, this does not necessarily mean more native species occurred. This only became evident once the study was conducted and was not the premise of the study rather it was what the study wanted to assess.
Results:
Capitalize only the first letter in family names.
Response: Table column modified
You note that you surveyed woody plants, but then report 6 grass species. Please clarify.
Response: line 128-130 of methods explains that grasses were also sampled
Most of the report of species numbers per family is redundant, having Table 1.
Response: Done in order to synthesize the table without having to look at it
Table 1: Add for each species at which site/s it was found.
Response: added column for site occurred
I am unsatisfied by the rarefaction analysis. The x-axis needs to be quantitative and continuous (as much as possible), especially if there are differences in area size among zones and/or sites.
Response: the sites were distinguished by zone and only 3 zones were established at each site.
Figure 1: Describing which curve relates to which case in both a textual and a graphical legend is redundant. Let alone that one river is not shown in the graphical legend Also the / one river is not shown in the graphical legend.
Response: Graphical legend extended to show 3 rivers
Also, the numbers do not match the text (20, 15 and 12 compared to 13, 25 and 18).
Response: added “for woody vegetation”
The authors sometimes refer to Dmg as “species richness index”, which is confusing and misleading.
Response: changed line 114 to “The Margalef’s index for species richness (Dmg)”, that was the only line the index was referred to as mentioned by reviewer
Section 3.3: You should show, possibly in a table, which differences are significant and which are not.
Response: suggest being added as an appendix
Figures 3+4: It is important to indicate significant differences within the graphs.
Response: suggest being added as an appendix
Section 3.6: I do not understand the rationale behind this analysis. Should it not be at the zone scale?
What information does it provide?
Response: this gives an understanding of the sampled riparian zone after collating the data
Tables 2+3: Why not have two smaller tables (one per index), directly comparing all three rivers or all three zones?
Response: this layout shows the difference between doing a site vs a zonal comparison
Discussion:
The first paragraph/sentence is redundant. I advise the authors to also read the following:
Katz O, Stavi I. (in press). Hierarchical effects of Tamarix aphylla afforestation in a sand dune environment on vegetation structure and plant diversity. Forest Science,
https://doi.org/10.1093/forsci/fxaa011
Response: kept to emphasise what the discussion while address, i.e. how biodiversity has been affection by exotic Tamarix invasion
Page 11, second paragraph: This is where an external control (Tamarix-free river) would have been very helpful. As it sits now, it cannot be 100% sure that the plant diversity decrease in zone 1 is
entirely related to the invasion. Page 11, last paragraph + page 12, first paragraph: The occurrences of T. usneiodes and A. muelleri could have been provided in the methods section, instead of misleading readers that all rivers are equal, and then surprising us.
Response: added “The Leeu River was the only site where the native congeneric, Tamarix usneoides, occurred.” Line
“This finding suggests that … most likely to establish”: Do you know that Tamarix was the first to establish and the others followed? Is it possible that this river was just more susceptible to invasions?
Response: This is true, however is known to promote the invasion of other invasive plants as it induces changes to the environment that are usually unsuitable for endemic species.
The following part about climate change is farfetched. At the very list, give evidence/reference for such a change, preferably show its impact on the studied ecosystems.
Response: sentence removed
Page 12 onwards: Most of this is too speculative, or at the very least not strongly connected to the main story and evidence and not leading anywhere clearly. The section referring to ecosystem services is important, but not well established. I very much liked the idea of having one, but
disliked this particular one.
Response: section highlights some of the ways that the biodiversity loss can affect ecosystem functions and services with the knowledge of the mentioned plant characteristics

Round 2
Reviewer 2 Report
I am satisfied with the revisions.
The authors have, without a doubt, considered all comments carefully and made the needed revisions.
Although I am not a native English speaker, I think that a minor English editing is required, but can probably be done during production.